# Analysis of Antibiotic Resistance Genes (ARGs) across Diverse Bacterial Species in Shrimp Aquaculture

**DOI:** 10.3390/antibiotics13090825

**Published:** 2024-08-30

**Authors:** Tilden M. Mitchell, Tin Ho, Liseth Salinas, Thomas VanderYacht, Nikolina Walas, Gabriel Trueba, Jay P. Graham

**Affiliations:** 1School of Public Health, University of California, Berkeley, CA 94704, USA; tilden_remerleitch@berkeley.edu (T.M.M.); tin@berkeley.edu (T.H.); tjvy@berkeley.edu (T.V.); nwalas@berkeley.edu (N.W.); 2Instituto de Microbiología, Colegio de Ciencias Biológicas y Ambientales, Universidad San Francisco de Quito USFQ, Quito 170901, Pichincha, Ecuador; lsalinast1@usfq.edu.ec (L.S.); gtrueba@usfq.edu.ec (G.T.)

**Keywords:** shrimp, AMR, aquaculture, ceftriaxone antibiotic resistance, 3GCR, multi-drug resistance, *E. coli*, genomic analysis, public health

## Abstract

There is little information available on antibiotic resistance (ABR) within shrimp aquaculture environments. The aim of this study was to investigate the presence of antibiotic resistance genes (ARGs) in shrimp farming operations in Atacames, Ecuador. Water samples (*n* = 162) and shrimp samples (*n* = 54) were collected from three shrimp farming operations. Samples were cultured and a subset of isolates that grew in the presence of ceftriaxone, a third-generation cephalosporin, were analyzed using whole-genome sequencing (WGS). Among the sequenced isolates (*n* = 44), 73% of the isolates contained at least one ARG and the average number of ARGs per isolate was two, with a median of 3.5 ARGs. Antibiotic resistance genes that confer resistance to the β-lactam class of antibiotics were observed in 65% of the sequenced isolates from water (20/31) and 54% of the isolates from shrimp (7/13). We identified 61 different ARGs across the 44 sequenced isolates, which conferred resistance to nine antibiotic classes. Over half of all sequenced isolates (59%, *n* = 26) carried ARGs that confer resistance to more than one class of antibiotics. ARGs for certain antibiotic classes were more common, including beta-lactams (26 ARGs); aminoglycosides (11 ARGs); chloramphenicol (three ARGs); and trimethoprim (four ARGs). Sequenced isolates consisted of a diverse array of bacterial orders and species, including *Escherichia coli* (48%), *Klebsiella pneumoniae* (7%), *Aeromonadales* (7%), *Pseudomonadales* (16%), *Enterobacter cloacae* (2%), and *Citrobacter freundii* (2%). Many ARGs were shared across diverse species, underscoring the risk of horizontal gene transfer in these environments. This study indicated the widespread presence of extended-spectrum β-lactamase (ESBL) genes in shrimp aquaculture, including *bla*_CTX-M_, *bla*_SHV_, and *bla*_TEM_ genes. Increased antibiotic resistance surveillance of shrimp farms and identification of aquaculture operation-level risk factors, such as antibiotic use, will likely be important for mitigating the spread of ARGs of clinical significance.

## 1. Introduction

Antibiotic resistance (ABR) represents a serious global threat with far-reaching economic and public health consequences [1]. A recent study estimated that 1.27 million deaths are attributed to ABR across the globe, and most of these occur in low- and middle-income countries [2]. To date, most research on ABR has focused on healthcare facilities, which are a critical element to the ABR crisis. A growing body of research, however, is highlighting the ways that exposures outside of healthcare facilities may also contribute to ABR.

Overuse of antibiotics in the farmed shrimp sector is a significant yet critically understudied area [3,4,5], with most research focused on South Asia [6,7] and East Asia [8,9]. Within the global seafood market, shrimp are a leading commodity, with an increasing proportion of their production stemming from intensive farming, which utilizes high stocking densities, manufactured feeds, and is associated with an increased risk of disease susceptibility [3,10]. The spread of ABR through aquatic environments is of particular concern because shrimp farms receive a variety of anthropogenic inputs such as human and livestock wastewater and stormwater runoff which can assist in the dissemination of ABR [11]. Shrimp aquaculture sites also foster new routes for human exposure to ABR via handling and consumption of contaminated shrimp; recreation in waters affected by the effluent from shrimp farms; occupational exposures; irrigation of nearby food crops with effluent from shrimp farms; and contamination of drinking water [12]. The global export of shrimp contaminated with drug-resistant bacteria and antibiotic resistance genes (ARGs) likely facilitates the dissemination of ABR [13,14].

According to the Food and Agricultural Organization of the United Nations, in 2022, Ecuador was the world’s leading producer of farmed shrimp, having harvested 1.3 million tons of shrimp that year, surpassing its previous production quantity in 2021 by 28.9 percent [15]. In the Western Hemisphere, Ecuador contributes 63.8% of the production of white-leg shrimp (*Litopenaeus vannamei*) [16]. The province of Esmeraldas, located on Ecuador’s northwestern coast, is a major producer of farmed shrimp, as its topography includes many bays, island archipelagos, and extensive mangrove forests.

The extent and magnitude of aquaculture’s contribution to the overall burden of ABR is poorly understood [6]. Recent studies in India, China, and Singapore have provided evidence that shrimp aquaculture environments can act as reservoirs of multi-drug-resistant *Escherichia coli* [6,8,17], yet very little data are available, particularly within the South American context [18,19,20]. Outside of the US and EU, only nine countries routinely publish national reports on antibiotic use and resistance, and there are no universally accepted policies governing the use of antibiotics in aquaculture systems [21].

Ecuador is a top producer of farmed shrimp [22]. There are approximately 2712 shrimp farms consisting of 1738 square kilometers in the Ecuadorian coastal provinces, El Oro, Esmeraldas, Guayas, Manabi, and Santa Elena [22]. Some research suggests that antibiotics are not used in Ecuadorian shrimp aquaculture [23] for biosafety reasons and due to environmental legislation, yet in 2024, the Ministerio de Produccion Comercio Exterior, Inverssiones y Pesca still lists several classes of antibiotics as acceptable for use in aquaculture, including tetracyclines and amphenicols [24]. Another factor potentially affecting antibiotic resistance in Ecuadorian aquaculture is the general lack of wastewater collection and treatment in the country, which can affect surface water quality and lead to antibiotics and ABR in aquaculture systems [25].

In 2024, the World Health Organization (WHO) released an updated List of Medically Important Antimicrobials as a guide for international antimicrobial stewardship [26]. This new categorization of antimicrobials advocates for a One Health perspective, recognizing that when categorizing the importance of antimicrobial agents, greater significance should be placed on antimicrobials used in animal production settings that can be transmitted to humans, as these cases present important opportunities for improved risk management strategies for antimicrobials in the animal sector [26]. Currently, there are no studies in Ecuador, to our knowledge, that have measured shrimp aquaculture as a potential reservoir of clinically important antibiotic-resistant bacterial species. This study highlights the importance of the shrimp farming environment as a potential site where ARG transfer may occur among a diverse array of clinically important bacterial species.

## 2. Results

### 2.1. Prevalence of Escherichia coli

In total, 216 samples were collected from three distinct shrimp farming operations in Atacames, Ecuador, designated as farms A, B, and C (Figure 1). These samples included 162 water samples from different shrimp farm sources: influent (*n* = 54), effluent (*n* = 54), and shrimp ponds (*n* = 54), as well as 54 shrimp samples taken at weekly intervals.

We calculated *E. coli* colony-forming units (CFUs) for 81 water samples (cultured without ceftriaxone) where the number of CFUs could be observed and counted, in order to assess fecal indicator loads (Table 1). Mean *E. coli* colony-forming units per 10 mL (EC cfu/10 mL) ranged from 27.8 (Farm A influent) to 66.3 (Farm A effluent). Median EC cfu/10 mL ranged from 7 (Farm A influent) to 77 (Farm A effluent). In every farm, the mean and median EC cfu/10 mL increased between where the water entered the shrimp farm (i.e., influent) and where the water left the shrimp farm (i.e., effluent).

For 106 samples (79/162 water samples and 27/54 shrimp samples), the antibiotic ceftriaxone was added to the culture media to select for bacterial isolates that grew in the presence of this third-generation cephalosporin (3GC). Nearly all of the water samples showed bacterial growth in the test kit bags with the ceftriaxone supplement added (Table 1 and Table 2).

We also sampled shrimp from each shrimp farm on a weekly basis during a three-week period in June 2023 (Table 2). Among shrimp gut rinse samples cultured in the presence of ceftriaxone, 67% to 78% of the samples showed growth in the presence of this 3GC. Shrimp Farm A showed 67% of samples positive for ceftriaxone-resistant bacteria, Shrimp Farm B showed 78% of samples positive for ceftriaxone-resistant bacteria, and Shrimp Farm C showed 67% of samples positive for ceftriaxone-resistant bacteria (Table 2).

### 2.2. Antibiotic Resistance Genes

Whole-genome sequencing (WGS) was conducted from a random subset of bacterial isolates (*n* = 44) that grew in the presence of ceftriaxone. Bacterial isolates were selected from water (*n* = 31) and shrimp samples (*n* = 13) for WGS. Among the 44 isolates sequenced, the average number of ARGs per isolate was two genes, with a median of 3.5 ARGs.

Sequencing revealed that many ARGs were shared between diverse bacterial species from the water and shrimp samples. ARGs that confer resistance to the beta-lactam class of antibiotics (*amp*H, *cph*A, *bla*_ACT_, and ESBL genes such as *bla*_CMH_, *bla*_CMY_, *bla*_CTX_, *bla*_OXA_, *bla*_PAO_, *bla*_SED_, *bla*_SHV_, *bla*_TEM_, additional details below) were observed in 65% (*n* = 20) of the isolates from water and 54% (*n* = 7) of the isolates from shrimp. ARGs that confer resistance to the following antibiotic classes were common: (1) macrolide genes (*mdf*A and *mph*A) were observed in 55% (*n* = 17) of water isolates and 38% (*n* = 5) of shrimp isolates; (2) tetracycline genes (*tet*A, *tet*B) were observed in 11% (*n* = 11) of water isolates and 13% (*n* = 1) of shrimp isolates; (3) aminoglycoside genes (*aac*, *aad*, *ant*, *aph*) were observed in 31% (*n* = 10) of water isolates and 8% (*n* = 1) of shrimp isolates; (4) fosfomycin gene (*fos*A) was observed in 23% (*n* = 7) of water isolates and 8% (*n* = 1) of shrimp isolates; (5) quinolone genes (*oqx*A, *oqx*B, *qnr*B, *qnr*S) were observed in 13% (*n* = 3) of water isolates and 23% (*n* = 4) of shrimp isolates. ARGs that confer resistance to the following classes were only seen in water isolates: (1) chloramphenicol genes were found in 26% (*n* = 8) of water isolates; (2) sulfonamide genes (*cat*B, *cml*A, *flo*R) were found in 26% (*n* = 8) of water isolates; (3) the trimethoprim gene (*dfr*A) was present in 10% (*n* = 3) of water isolates. Over half of all the isolates (59%, *n* = 26) carried ARGs that confer resistance to more than one class of antibiotics (Figure 2).

### 2.3. Bacterial Species

A wide array of bacterial species was also present in the sequence isolates (Figure 3), although the Aquagenx^®^ water quality test kits used in this study were designed for selectively culturing *E. coli*. Within the bacterial genomes sequenced, four species fell within the order *Enterobacterales*: *Escherichia coli* (48%), *Klebsiella pneumoniae* (7%), *Enterobacter cloacae* (2%), and *Citrobacter freundii* (2%). The orders *Pseudomonadales*: *Pseudomonas putida* (11%), *Pseudomonas aeruginosa* (5%), and *Aeromonadales* (7%) were the other Gammaproteobacteria found. In eight (18%) of the bacterial sequences, the species were unidentified.

A large number of ARGs were identified among the isolates sequenced (Figure 4). Overall, we found 61 different antibiotic resistance genes across the 44 sequenced isolates (see Appendix A). The 61 ARGs identified in the isolates confer resistance to nine antibiotic classes. ARGs for certain antibiotic classes were more common, including beta-lactams (26 ARGs); aminoglycosides (11 ARGs); chloramphenicol (three ARGs); trimethoprim (four ARGs); fosfomycin (six ARGs); macrolides (two ARGs); quinolones (five ARGs); sulfonamides (two ARGs); and tetracyclines (three ARGs). The most prevalent ARG was *mdf*A, conferring resistance to macrolide–lincosamide–streptogramin B antibiotics, and it was detected across 26 isolates. Other ARGs with notable frequencies included *tet*A genes (*n* = 6). Twenty-six different genes associated with beta-lactam antibiotic resistance were seen across the sequenced isolates. A high number of isolates (*n* = 18) harbored *bla*_CTX-M_ genes, and many of those (*n* = 11) were the *bla*_CTX-M-8_ gene. Additionally, the *oqx*B and *oqx*A genes (*n* = 6 each) were found and are related to reduced fluoroquinolone susceptibility and resistance to multiple antibiotics.

The MOB-suite [27,28] plasmid reconstruction and typing tool was utilized, and we found that 16 of our 44 sequences (36%) contained plasmids that carried a total of nine ESBL genes, consisting of different allelic variants of *bla*_CTX-M_, *bla*_CMY_, *bla*_OXA_, and *bla*_TEM_.

Many of the ARGs identified were found in multiple bacterial species (Figure 5). For example, the *E. coli* species shared ARGs across different species, including *Klebsiella pneumoniae*, *Enterobacter cloacae*, and *Pseudomonas putida.* The *mdf*A gene was identified across two species: *P. putida* and *E. coli*. In looking at the ARGs shared between *K. pneumoniae* and *E. coli*, we observed potential pathways for cross-species gene transfer of ceftriaxone resistance genes, such as the *bla*_TEM-1B_ and *bla*_CTX-M-15_, as well as a range of genes conferring resistance to aminioglycosides (*aph**), macrolides (*mph*A), sulfonamides (*sul**), tetracyclines (*tet*A), and trimethoprim (*dfr*A). Eight isolates, where the species could not be identified, were excluded from Figure 5.

### 2.4. Genotypes of Escherichia coli

Multilocus sequence typing (MLST) analyses revealed a diverse set of sequence types among the various species identified. Among the 21 *E. coli* isolates, we identified 12 sequence types: ST117, ST48, ST641, ST120, ST1204, ST155, ST2705, ST351, ST3580, ST48, ST6018, ST6186, ST641, and ST8612, in addition to one novel, unclassified sequence type. Of these, the first three STs have been identified previously as extraintestinal pathogenic *E. coli* (see Section 3).

Among the three *K. pneumoniae* isolates, we identified three sequence types: ST15, ST1310, and ST661. Out of these, ST15 has been previously associated with disease. Among the *P. aeruginosa* isolates, we identified two sequence types: ST2629 and ST1414, neither of which have been described in infections. Finally, for *P. putida*, ST3 and an unknown ST were found.

## 3. Discussion

Our investigation of antibiotic-resistant bacteria within the shrimp aquaculture environment of Ecuador found that most water and shrimp samples were positive for ceftriaxone-resistant bacteria. The presence of such 3GCR bacteria identified in the water samples of the shrimp farm influent suggest that fecal contamination upstream of the farms may be an important source of these organisms (i.e., they are not coming solely from shrimp farming), which highlights the need for cleaner source water and likely more wastewater collection and treatment. Within the shrimp farms, a range of bacterial species shared diverse ARGs. These findings highlight the potential for antibiotic-resistant genes to transfer between diverse bacterial orders. The finding of ARGs being shared across a broad range of species suggests that using indicator bacteria only (e.g., *E. coli*) may not capture the diversity of resistance genes present in an ecosystem. In this study, however, we did observe that *E. coli* was common in all of the samples, and they shared ARGs with five different species of bacteria, which may indicate that it is an adequate indicator species for aquaculture environments. Alternatively, the Aquagenx^®^ water quality test kits were designed to enrich for *E. coli*, which increased the probability that we identified *E. coli* in our samples. Although many of the bacteria detected in this study may not be pathogenic, the majority of the isolates identified carried important ARGs that can potentially be transmitted horizontally to pathogens either in human intestines or outside the host. While this study did not conduct phenotypic validation of these resistance genes (due to a lack of funding), their presence nonetheless poses serious implications for the spread of ABR across the food chain. Bacterial isolates from both shrimp and water carried ARGs that can confer resistance to over nine antibiotic classes, many of which fall under the WHO’s “List of Medically Important Antimicrobials”, which pose serious concern for human health [26].

Our study highlighted the presence of ARGs that confer resistance to WHO’s Highest priority critically important antimicrobials, which are compounds authorized for treatment of infections in both animals and humans [26]. Among the isolates that were sequenced from our samples of water and shrimp, beta-lactam resistance genes were commonly found, including 54% (*n* = 20) of the isolates from water and 65% (*n* = 5) of the isolates from shrimp. This includes the resistance to third-generation cephalosporins with the presence of *bla*_CTX-M_, *bla*_SHV_, *bla*_OXA_, and *bla*_TEM_ genes. The treatment of infections can be complicated if these ARGs are present in the isolates causing the infection. Our study also found *bla*_OXA-486_, a gene that confers resistance to carbapenems, a last-line antibiotic for treating serious infections in hospitals. A limitation of this study was the lack of phenotypic testing; focusing only on the presence of antibiotic resistance genes, as this study did, may result in false-positive predictions of phenotypic resistance. Thus, future studies should include antibiotic susceptibility testing.

The sharing of ARGs across diverse species was an important finding, highlighting the potential that ARGs can spread horizontally across a broad range of organisms, making controlling and managing the spread of ABR more challenging [29,30,31]. Among these, members of the order *Enterobacterales* stand out, given their significant capacity for harboring and transmitting ARGs [18,32,33,34]. Compared to fish species, crustacean farming presents distinct challenges and risks for ABR, primarily because shrimp lack an acquired immune system, which increases their susceptibility to pathogens and the need for antimicrobial treatments [12,19,29]. Additionally, unlike other fish industries that have seen a reduction in localized disease outbreaks in farmed fish species due to vaccination, shrimp have a more primitive immune system that does not respond to vaccination [3]. There is evidence demonstrating that horizontal gene transfer can take place between unrelated bacterial species, including non-pathogenic environmental bacteria, leading to widespread resistance to a variety of antibiotics across many types of bacteria [35,36]. It is important to note that there are no antibiotics that have been developed solely for use in shrimp, therefore the antibiotics used in shrimp production are the same antibiotics licensed for the treatment of human infections [37].

Environmental factors, such as the presence of heavy metals, herbicides, and biocides, can affect ARG expression, and biofilms provide a niche for diverse microbes and serve as potential spaces for horizontal gene transfer [38,39,40]. Studies have shown that even minimal antibiotic concentrations can lead to the evolution of distinct resistance mechanisms, suggesting that low-level selective pressures can drive the development of significant resistance [41].

Khan et al. studied ABR in shrimp farms in Bangladesh and found that the majority of the bacteria were susceptible to ciprofloxacin; only *Proteus alimentorum* was weakly resistant [7]. In our samples, however, ARGs that confer resistance to the quinolone class of antibiotics were observed in 13% (*n* = 3) of the water isolates and 23% (*n* = 4) of the shrimp isolates. The quinolones class, which includes fluoroquinolones such as ciprofloxacin, are effective drugs used to counter a broad spectrum of bacterial infections. Ciprofloxacin has been found to be broadly applied in shrimp aquaculture to manage infections caused by Gram-negative bacteria, such as *Pseudomonas* spp., and on occasion employed to treat infections caused by Gram-positive bacteria [42]. Unlike the Khan et al. study, our results showed that quinolone resistance genes were present in the shrimp aquatic ecosystems.

A 2022 study analyzing water bodies receiving hospital effluents in Kerala, India, highlighted the prevalence of ESBL-producing *E. coli* and *K. pneumoniae* with high rates of multi-drug resistance that were confirmed to have transferred resistance genes via plasmid-mediated transfer [43]. The role of *K. pneumoniae* in ABR and a One Health context is well established, and it is a species known for serving as a reservoir for a wide array of ARGs. Its significant genetic diversity and common carriage of plasmids are thought to facilitate the transfer of resistance traits to various pathogens, including key carbapenemase genes [44]. Our research aligns with this perspective, demonstrating shared ARGS in both *K. pneumoniae* and *E. coli* (e.g., *bla*_CTX-M-15_, *bla*_TEM-1B_, *mph*A, *tet*A, and *sul*_1-2_).

Recent findings highlight the public health risks associated with the transfer of ARGs in *Aeromonas* spp. found in water bodies [31]. This species, which has been shown to cause severe illnesses such as gastroenteritis and septicemia, has a propensity to acquire and disseminate ARGs, most notably through mechanisms like conjugation [31,45]. In our sample, both *Aeromonas* spp. and *E. coli* had genes that produce CTX enzymes (*bla*_CMY-8B-1_ and *bla*_CMY-2-1_). Whether one is the source of the ARG for the other was not characterized in this current study; however, it points to a concerning pathway for the spread of beta-lactamase resistance genes. Compounding this concern, the detection of the *oqx*A_1 and *oqx*B_1 genes in *E. cloacae*, *E. coli*, and *K. pneumoniae* within our samples underscores that gene transfer could be an important mechanism in aquaculture environments.

The MLST analysis of the 44 isolates highlighted a diverse array of sequence types across various species but also demonstrated that several sequence types were shared among both shrimp and water isolates. In a study conducted by Zhang et al. in 2024, they found a multi-drug-resistant *E. coli* ST2705 in retail fish products in China [46]. This study was the first report of *mcr-1*-positive ESBL-producing *E. coli* ST2705 and ST10 in retail fish, and while our study did not contain any *mcr-1*-positive isolates, we did identify *E. coli* ST2705, which has also been identified in livestock and flies [47]. Another clinically relevant sequence type found in our isolates was ST48, which has been shown to harbor multiple conjugative plasmids containing numerous ARGs [48,49].

Across our isolates, we identified *E. coli* ST641 and ST6186 in both shrimp and water samples. In a study by Do et al. (2023), ST641 was observed in both human and swine isolates and demonstrated closely related phylogenetic connections, including similar virulence characteristics [50]. Additionally, sequence types for other species like *K. pneumoniae* (e.g., ST15, ST661, and ST1310) and *P. aeruginosa* (e.g., ST2629 and ST1414) were also identified. Among these, ST15 in *K. pneumoniae* stands out as a linage that has been found to cause clinical infections; it is known as an opportunistic pathogen that can cause pneumonia, septicemia, bronchitis, and urinary tract infections (UTIs), and it has a multi-drug-resistant lineage [51].

The UN Food and Agriculture Organization has identified food consumption as “likely to be quantitatively the most important potential transmission pathway from livestock to humans, although direct evidence linking ABR emergence in humans to food consumption is lacking” [37]. In Ecuador, the consumption of street food is widespread and an understanding of risks from foodborne pathogens and antimicrobial resistance is limited [18].

In the last decade, there has only been one study conducted in Ecuador that examined ABR in shrimp farms and retail shrimp [20]. Sperling et al. primarily focused on the prevalence of *Vibrio* spp. (95.6%) in shrimp and found that many *V. parahaemolyticus* strains were multi-drug-resistant. Our study greatly expands these previous findings by providing a broader perspective of the diverse species and ARGs in the aquaculture environments of Ecuador.

Notwithstanding the limited number of shrimp facilities examined in this study, these results underscore important implications for environmental and human health. Ecuadorian aquaculture practices directly expose shrimp farmworkers and the surrounding ecosystem, including extensive protected mangrove forests, to drug-resistant bacteria [52,53]. Furthermore, the global market for Ecuadorian shrimp is at a current all-time high, including a new trade deal with China that included 4 billion USD in shrimp sales in 2022 [54]. With high levels of international trade within the shrimp industry, the extent to which the trade and consumption of shrimp plays a role in the global dissemination of ABR is still unknown [3]. According to Romo-Castro, Ecuador does not yet adhere to WHO antibiotic stewardship strategies, and data including basic metrics of antibiotic use in agriculture and aquaculture are still lacking [19]. Lastly, there are no standardized international guidelines at present for maximum antimicrobial residue limits in water. Water therefore becomes an important vehicle for the spread of both antimicrobial residues and resistance, since contaminated water can be used for other agricultural purposes or consumed directly by humans and livestock.

The results generated from this study increase our understanding of the factors that are associated with the spread of ABR through aquaculture systems. Despite the relatively low number of isolates that were sequenced (*n* = 44), the results of this study draw attention to the fact that shrimp farms in Atacames, Ecuador may potentially be key sites for the sharing of resistance genes across clinically significant species, which can subsequently have negative impacts on consumers’ health. The threat of antibiotic resistance is a growing cause for concern and more effort is needed to prevent the emergence of new resistant strains and the spread of existing ones to humans.

## 4. Potential Future Research

There is a significant risk of ABR dissemination associated with shrimp production and new efforts are needed to reduce the development and spread of ABR from shrimp ponds to other environments. The first step that is needed is to move towards more judicious use of antibiotics in the shrimp farming process [55]. More studies should adopt a One Health approach to sampling methods that focuses on all aspects of the shrimp-rearing environment and surrounding ecosystem, including but not limited to shrimp pond sediment, shrimp pond water at various stages of shrimp development, and adjacent aquatic systems [56].

Public health researchers and practitioners have advocated for a global monitoring system for ABR surveillance [55]. Global monitoring of ABR is not only important to understand the total burden of ABR dissemination but is also important to identify countries which could benefit from targeted antibiotic-use reduction initiatives. It is important to note, however, that increasing laboratory facilities and equipment for genomic and phenotypic testing of isolates remains a major hurdle for many countries [57]. The top ten shrimp-producing countries do have regulations in place regarding the use of antibiotics; however, it is crucial to note that there is a major difference between regulation and enforcement. Another key way to promote the judicious use of antibiotics in the shrimp-rearing process is changing the type and quantity of feed used in the process. Using feed with antibiotic additives is a widespread practice across the industry, and this practice should be stopped [55].

There are several key mitigation strategies that could be implemented immediately among shrimp producers that would greatly decrease the spread of ABR. First, antibiotic stewardship needs to be improved and could include the administration of antibiotics only under the supervision of a veterinarian. Alternatives to antibiotics in the shrimp industry are growing, and there are several that are currently under development. These strategies include vaccine development, phage therapy, quorum quenching, and probiotics [55,57]. There is increasing scientific evidence pointing to the potential benefits and use of innate immune memory in shrimp, which could offer possibilities for vaccination [58].

## 5. Materials and Methods

### 5.1. Study Area

The study was conducted in June 2023 in Atacames, Ecuador, a coastal town with a population of 51,204 according to the last national census in 2022 [59]. The site was selected due to the scale of shrimp production in the area; Ecuador exported over one million tons of shrimp to the global market in 2022 [15].

A convenience sampling method was used to identify appropriate shrimp farms for sampling using the following criteria: (1) accessibility to water influent and effluent channels and ponds within shrimp aquaculture facilities; (2) availability of commercial shrimp for sampling; and (3) proximity of the shrimp farm to populated residential areas.

### 5.2. Water Sample Collection and Analysis

Surface water samples were collected aseptically from three distinct aquaculture farms (*n* = 3), and a total of 162 surface water samples were taken from the farm influent (*n* = 54), effluent (*n* = 54), and within the shrimp farm ponds (*n* = 54). Eighteen surface water samples were collected on a weekly basis from each of the three shrimp aquaculture facilities.

Water samples were collected and processed using the Aquagenx portable water quality test kits (Aquagenx, Chapel Hill, NC, USA) according to the manufacturer’s guidelines [60]. Given the high levels of *E. coli* counts in the surface water samples, a 1:10 dilution was made, and 10 mL of the surface water was diluted with 90 mL of autoclaved, distilled water. Half of all surface water samples (*n* = 81) were spiked with 1 mg/L ceftriaxone for further analysis. The samples were then incubated at an ambient temperature (24 °C) for 48 h.

### 5.3. Shrimp Sample Collection and Analysis

Shrimp samples were procured directly from designated shrimp ponds from each of the 3 shrimp farms at weekly intervals for 3 weeks, resulting in a total of 54 shrimp samples. Every week, eight samples were collected, comprising three replicates to detect and quantify *E. coli* and an equal number of samples supplemented with ceftriaxone. One negative and one positive control sample were run with each batch of samples. Shrimp samples were placed in Whirl-Pak^®^ Bags (Whirl-Pak, Madison, WI, USA) and transported back to the lab on ice for further analysis.

Each shrimp sample involved the aseptic removal of the carapace and the extraction of the shrimp intestine. The shrimp intestine was then placed in a whirl pack bag along with 100 mL of autoclaved distilled water. The shrimp and water combination (shrimp rinse) was then shaken for one minute, and 100 mL was aseptically transferred into an Aquagenx water quality test bag. Half of the shrimp samples were spiked with 1 mg/L ceftriaxone (Daehwa Pharmaceuticals, Seoul, Republic of Korea).

### 5.4. Isolation Procedure for Escherichia coli

In the case of the water and shrimp samples subjected to ceftriaxone exposure, an initial quantification of *E. coli* was performed. Subsequently, from the gel sample bags, three bacterial isolates that grew in the presence of ceftriaxone (i.e., a third-generation cephalosporin) were randomly selected and stored. The isolates were preserved by placing them in a 15% glycerol solution (Sigma-Aldrich, Burlington, MA, USA) and freezing them at a temperature of −20 °C. All preserved isolates were packed on ice and transported to the Instituto de Microbiología at Universidad de San Francisco de Quito (USFQ), located in Cumbaya.

This study used Aquagenx portable water quality test kits (Aquagenx, Chapel Hill, NC, USA) for the identification of *E. coli* CFU counts in both water and shrimp samples. The Aquagenx portable water quality test kits are designed to only grow *E. coli*; however, studies have found that other species can grow in these kits [61].

### 5.5. DNA Sequence and Analysis

Bacterial isolates that grew in the presence of ceftriaxone, a third-generation cephalosporin (3GC), were plated on MacConkey lactose agar (Dipco, Sparks, MD, USA) supplemented with ceftriaxone (1 mg/L) and incubated at 37 °C for 24 h. Next, one colony was selected and cultured in Tryptic Soy Broth (TB) (Dipco, Sparks, MD, USA) at 37 °C for 16 h. Genomic DNA from the isolates was extracted using the DNeasy Blood and Tissue Kit (QIAGEN, Hilden, Germany) according to the manufacturer’s instructions.

Sequencing was carried out by Novogene (Sacramento, CA, USA) using a single 2 × 250-bp dual-index run on an Illumina (San Diego, CA, USA) MiSeq with Nextera XT libraries to generate ~30- to 50-fold coverage per genome. Genome assembly of MiSeq reads for each sample was performed using Unicycler version 0.5.0 with all default parameters [62], which invoked the SPAdes assembler with automated k-mer detection [63]. The identification of the genus and species of the isolates was carried out using Seemann’s MLST version 2.19.0 [64], using the database from PubMLST [65]. ARGs were identified using the ABRicate tool (version 1.0.1) [66], and Resfinder was the database used, with >80% as threshold for the identification of resistance genes [67]. Additionally, multilocus sequence typing (MLST) was conducted in accordance with species-specific PubMLST allele schemes [68,69]. Detection of mobile genetic elements, such as plasmids, was carried out using the MOB-suite version 3.1.9 with its default parameters [27,28]. Data processing programs were run in a Linux High Performance Cluster provided by the University of California, Berkeley, with parallelization assisted by use of the GNU Parallel version20190922 [70]. Raw reads from isolates sequenced in this study are available at the NCBI Short Read Archive (SRA).

## 6. Conclusions

In this study, we found that shrimp farms were a key reservoir of ARGs across diverse bacterial species. This poses a significant threat to both public health and aquatic environments. Based on the study findings, we suggest that further environmental monitoring of ABR across aquatic systems and the shrimp supply should take place, including the influent water quality of these aquaculture systems. This increased surveillance could provide further insights into how to better safeguard the food supply and human health. The results of this study illustrate the need for more transparent accounting of antibiotic use by shrimp producers. Antibiotic stewardship in this sector could go a long way towards improving the safety of the food supply and protecting human health.

## Figures and Tables

**Figure 1 antibiotics-13-00825-f001:**
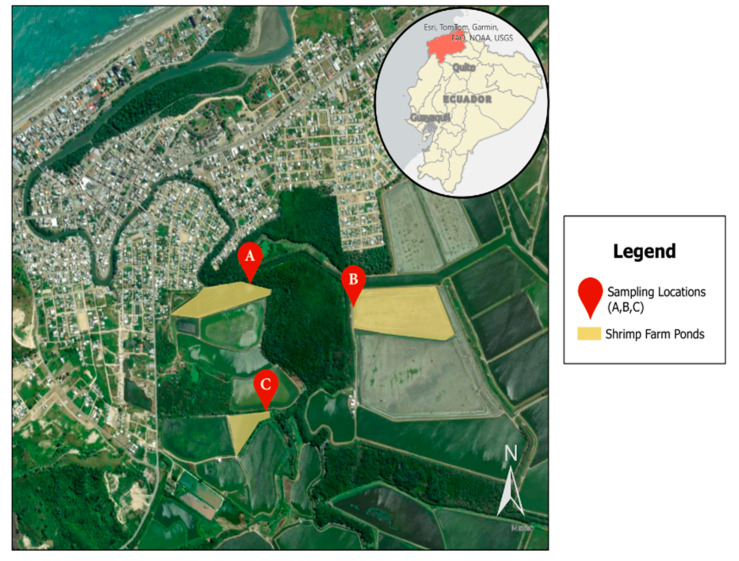
Study sampling locations in Atacames, Ecuador, located in the northwestern province of Esmeraldas. Three sampling locations were included that all connect to the Atacames River.

**Figure 2 antibiotics-13-00825-f002:**
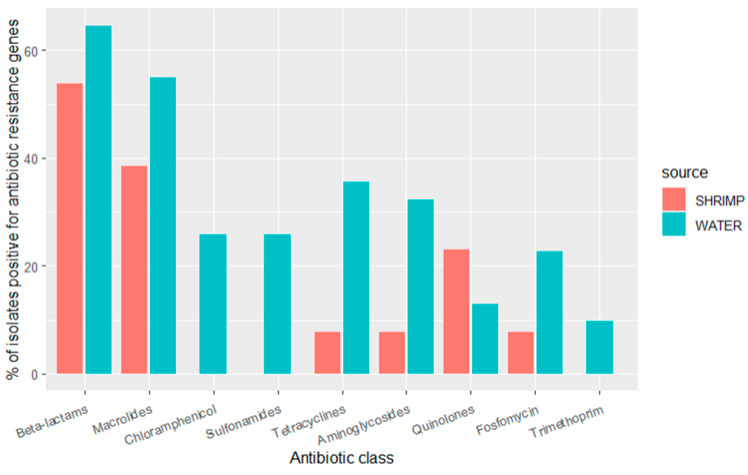
Prevalence of antibiotic resistance genes by antibiotic class for isolates detected in water samples (*n* = 31 isolates) and shrimp samples (*n* = 13 isolates) in shrimp farms.

**Figure 3 antibiotics-13-00825-f003:**
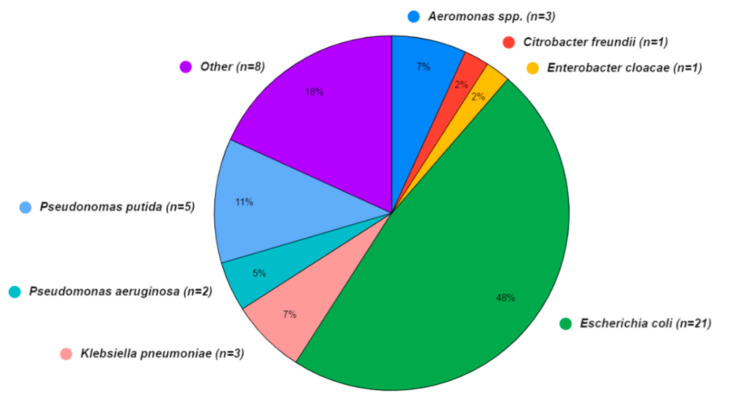
Bacterial species identified in water (*n* = 31) and shrimp samples (*n* = 13) from shrimp aquaculture farms in Ecuador.

**Figure 4 antibiotics-13-00825-f004:**
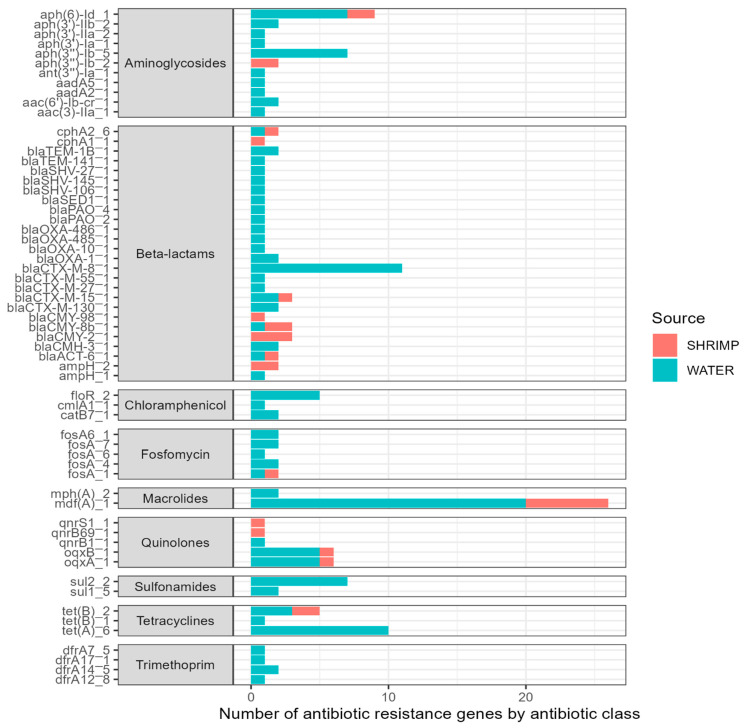
Type (*y*-axis) and number (*x*-axis) of antibiotic resistance genes (ARGs) among sequenced isolates (*n* = 44).

**Figure 5 antibiotics-13-00825-f005:**
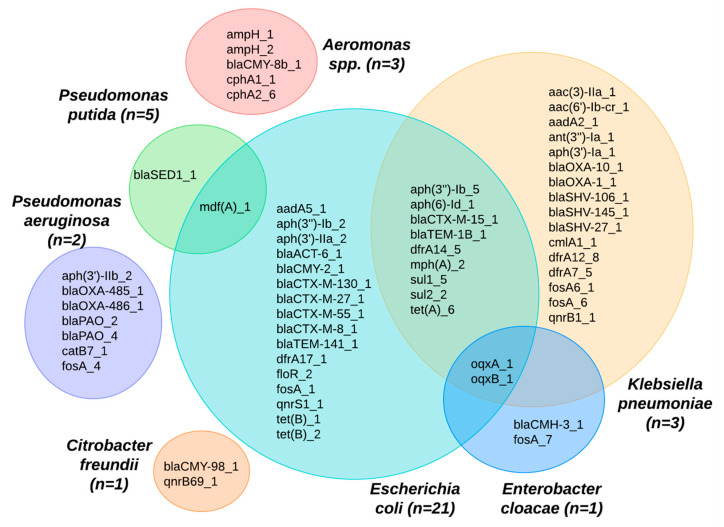
Venn diagram showing ARGs identified among different bacterial species from shrimp farms.

**Table 1 antibiotics-13-00825-t001:** Summary of water quality results indicating *E. coli* CFU counts and percentage of samples positive for growth in the presence of ceftriaxone across all sampling locations.

Sample Location	Sample Type	Number of Samples	Mean *E. coli* CFU (per 10 mL)	Median *E. coli* CFU (per 10 mL)	Percent of Samples with Bacterial Growth in Presence of Ceftriaxone
A—Influent	Water	18	27.8	7	100%
A—Effluent	Water	18	66.3	77	100%
A—Shrimp pond	Water	18	41.4	25	100%
B—Influent	Water	18	44	43	100%
B—Effluent	Water	18	54.8	56	100%
B—Shrimp pond	Water	18	48	61	100%
C—Influent	Water	18	39.4	20	89%
C—Effluent	Water	18	47.6	49	100%
C—Shrimp pond	Water	18	62.6	60	100%

**Table 2 antibiotics-13-00825-t002:** Summary of shrimp intestines results indicating the percentage of samples positive for bacterial growth in the presence of ceftriaxone across all sampling locations.

Sample Location	Sample Type	Number of Samples	Percent of Samples with Bacterial Growth in Presence of Ceftriaxone
A	Shrimp intestines	18	67%
B	Shrimp intestines	18	78%
C	Shrimp intestines	18	67%

## Data Availability

The data presented in this study are openly available at NCBI Short Read Archive (SRA) under BioProject ID PRJNA1151051 (http://www.ncbi.nlm.nih.gov/bioproject/1151051, accessed on 1 May 2024).

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
