# Peer review of "Analysis of Antibiotic Resistance Genes (ARGs) across Diverse Bacterial Species in Shrimp Aquaculture"

_antibiotics, 2024, doi:10.3390/antibiotics13090825_

Round 1
Reviewer 1 Report
Comments and Suggestions for Authors
In this manuscript, the authors investigate the types of antibiotic-resistant bacteria and antibiotic resistance genes (ARGs) in shrimp farming in Atacames, Ecuador. This research is of significant importance for mitigating the occurrence and spread of ARGs. The literature review is comprehensive, the results are clearly described, and the discussion is fairly thorough. However, there are still some issues that need to be addressed. I recommend reconsidering it for publication after the following revisions:
# Major Comments:
(1) Background Data on Environmental Factors: It is essential to add background data on environmental factors that could influence the study.
(2) Phenotypic Validation of Resistance Genes: The authors have used gene prediction to identify resistance genes and types. It is important to verify whether these genes actually confer the predicted resistance phenotypes. I recommend conducting phenotypic validation on the isolated strains.
(3) Location and Mobility of Resistance Genes: The manuscript can analyze the location of the identified resistance genes and their relationship with mobile genetic elements. This information is crucial for assessing the risk of resistance gene dissemination.
(4) Detailed Description of WGS Bioinformatics Analysis: The description of the bioinformatics analysis performed with whole-genome sequencing (WGS) is not specific enough. Parameters and thresholds used in tools like ABRicate for resistance gene identification should be detailed.
# Other Specific Suggestions:
(1) Line 32: The keyword "AMR" should be corrected to "ABR".
(2) Line 44: There is a cluster of references. It may be helpful to provide a brief summary of these references.
(3) Line 120: There is an extra space after "ARGs".
(4) Line 360: There is an extra period after the reference [52].
(5) Line 431 (Conclusion Section): Avoid citing references in the conclusion section.
(6) Research Methodology: It might be beneficial to discuss the strengths and weaknesses of the study’s methodologies. Consider mentioning the limitations of the isolation methods and suggest that combining them with more efficient techniques like metagenomics and binning could enhance the study.
Author Response
Reviewer #1
In this manuscript, the authors investigate the types of antibiotic-resistant bacteria and antibiotic resistance genes (ARGs) in shrimp farming in Atacames, Ecuador. This research is of significant importance for mitigating the occurrence and spread of ARGs. The literature review is comprehensive, the results are clearly described, and the discussion is fairly thorough. However, there are still some issues that need to be addressed. I recommend reconsidering it for publication after the following revisions:
(1) Background Data on Environmental Factors: It is essential to add background data on environmental factors that could influence the study.
We have now added additional background information on the environmental factors that could influence the study results (see lines 73-83).
(2) Phenotypic Validation of Resistance Genes: The authors have used gene prediction to identify resistance genes and types. It is important to verify whether these genes actually confer the predicted resistance phenotypes. I recommend conducting phenotypic validation on the isolated strains.
This is an important point, however, we are unable to go back to conduct phenotypic resistance analyses of the isolates (no more funding for this study). Screening for antibiotic resistance genes, as we did in this study can be associated with false-positive predictions of phenotypic resistance. Thus, we have now gone back to the discussion section to highlight this limitation (lines 244-247)
(3) Location and Mobility of Resistance Genes: The manuscript can analyze the location of the identified resistance genes and their relationship with mobile genetic elements. This information is crucial for assessing the risk of resistance gene dissemination.
TH: Given that we used selective media, which selects for third generation cephalosporin resistance, we have now analyzed the location of the ESBL genes and provided this new information within the manuscript.
(4) Detailed Description of WGS Bioinformatics Analysis: The description of the bioinformatics analysis performed with whole-genome sequencing (WGS) is not specific enough. Parameters and thresholds used in tools like ABRicate for resistance gene identification should be detailed.
TH: Thank you for this feedback. We have now added more detail to the methods about the parameters and thresholds used.
# Other Specific Suggestions:
(1) Line 32: The keyword "AMR" should be corrected to "ABR".
Thank you, this has now been changed.
(2) Line 44: There is a cluster of references. It may be helpful to provide a brief summary of these references.
Thanks for the feedback, the references are better explained now.
(3) Line 120: There is an extra space after "ARGs".
Fixed
(4) Line 360: There is an extra period after the reference [52].
Fixed
(5) Line 431 (Conclusion Section): Avoid citing references in the conclusion section.
We have now removed the citation from the concluding paragraph.
(6) Research Methodology: It might be beneficial to discuss the strengths and weaknesses of the study’s methodologies. Consider mentioning the limitations of the isolation methods and suggest that combining them with more efficient techniques like metagenomics and binning could enhance the study.
As the reviewer has noted, there are some important limitations that need more discussion. We have now added in additional limitations to using Aquagenx test kits, using ceftriaxone in our selective media and the lack of antibiotic susceptibility testing that could provide results on phenotypic resistance.
Reviewer 2 Report
Comments and Suggestions for Authors
The main aim of this study was to investigate the presence of antibiotic resistant bacterial species and antibiotic resistance genes (ARGs) in shrimp farming operations in Atacames, Ecuador.
The main issue of this article is related to the ARG detection because no information of percentage of identity of the ARG detected sequences was informed. In a supplementary file the authors must include the ARG sequences of isolates and their percentages of identity and accession numbers of both compared sequences (isolates from this study and Genbank sequences). It is not clear the level of identity considered for a positive detection of the ARGs.
Otherwise, it is difficult to understand why the authors not performed susceptibility assays of isolates to confirm that declared ARGs are able of conferring antimicrobial resistance.
The experimental design is very confusing. It is not clear the reason of applying an antimicrobial selective pressure using ceftriaxone, the used concentration (quite low), etc. Furthermore, no levels of antimicrobial resistant bacteria were determined, which could be interesting of performing to compare among three sampled shrimp farms as well as between influent and effluent samples.
The authors must provide information on the use of antimicrobial agents in sampled shrimp farms. In addition, the authors must comment antimicrobial agents approved for using in shrimp industry in Ecuador.
Lines 82-83: Paragraph “This study highlights the importance of the shrimp farming environment as a hub for ARG transfer among a diverse array of clinically important bacterial species” must be removed or changed because no ARG transfer assays were performed in the study.
In Figure 5 change “Spp.” to read “spp.”
Lunes 400-401: Paragraph “…cephalosporin) were randomly selected and stored…” The authors must describe if these strains were properly isolated describing used method and medium.
Lines 425-426: Paragraph “In this study, we found that shrimp farms were a key reservoir and pathway for the dissemination of ARGs across diverse bacterial species.” Must be changed or removed because no transfer assays or detection of mobilome elements were detected.
Comments on the Quality of English Language
Quality of English language is fine, not requiring major edition
Author Response
Reviewer #2
The main aim of this study was to investigate the presence of antibiotic resistant bacterial species and antibiotic resistance genes (ARGs) in shrimp farming operations in Atacames, Ecuador. The main issue of this article is related to the ARG detection because no information of percentage of identity of the ARG detected sequences was informed. In a supplementary file the authors must include the ARG sequences of isolates and their percentages of identity and accession numbers of both compared sequences (isolates from this study and Genbank sequences). It is not clear the level of identity considered for a positive detection of the ARGs. Otherwise, it is difficult to understand why the authors not performed susceptibility assays of isolates to confirm that declared ARGs are able of conferring antimicrobial resistance.
Thank you. We have now provided more detail about the parameters and thresholds for identifying ARGs.
The experimental design is very confusing. It is not clear the reason of applying an antimicrobial selective pressure using ceftriaxone, the used concentration (quite low), etc. Furthermore, no levels of antimicrobial resistant bacteria were determined, which could be interesting of performing to compare among three sampled shrimp farms as well as between influent and effluent samples.
This was a small exploratory study where there was limited resources and the scope of the work was limited by what was written in the original study proposal. We have now highlighted the limitations of the design and made additional clarifications in the methodology section.
The authors must provide information on the use of antimicrobial agents in sampled shrimp farms. In addition, the authors must comment antimicrobial agents approved for using in shrimp industry in Ecuador.
This is a good point, and we have now provided information on what antimicrobials are approved for the shrimp industry. However, the aquaculture industry is not amenable to sharing the specific information on antibiotic use given that this can have reputational consequences. Most shrimp farms did not want to participate in the research and getting information on antibiotic use was not possible.
Lines 82-83: Paragraph “This study highlights the importance of the shrimp farming environment as a hub for ARG transfer among a diverse array of clinically important bacterial species” must be removed or changed because no ARG transfer assays were performed in the study.
This is an important point and we have changed the text to reflect this comment. We have now changed the sentence to “This study highlights the importance of the shrimp farming environment as a potential site where ARG transfer may occur among a diverse array of clinically important bacterial species.”
In Figure 5 change “Spp.” to read “spp.”
Thanks for catching that. Fixed in the new diagram.
Lines 400-401: Paragraph “…cephalosporin) were randomly selected and stored…” The authors must describe if these strains were properly isolated describing used method and medium.
We have added additional details about the methods used for isolation, and additional references for the methods have been provided.
Lines 425-426: Paragraph “In this study, we found that shrimp farms were a key reservoir and pathway for the dissemination of ARGs across diverse bacterial species.” Must be changed or removed because no transfer assays or detection of mobilome elements were detected.
We have changed this to state, “In this study, we found that shrimp farms were a key reservoir of ARGs across diverse bacterial species”. Additionally, we added information in the text about plasmids content and ARGs contained in them.
Reviewer 3 Report
Comments and Suggestions for Authors
Review of the article Analysis of Antibiotic Resistance Genes (ARGs) Across Diverse Bacterial Species in Shrimp Aquaculture
This the review for the above mentioned article, that deals with ARGs in shrimp aquaculture located in the same area and all have connection to the same Atacames River.
The article is well written but there are some key elements that need to be clarified since there is some confusion with the sampling process and the presentation of the results.
In table 1, how many times the A was sampled? And why not the same for B and C?
In Table 2 there is a difference in results between A and B-C, has this been explained in article? Because their water samples are well infected
Line 117-120. WGS was conducted to a “random” subset of isolates. Why was that? And why the majority was chosen from water samples 34 and only 10 from shrimp samples?
Line142-148. Aquagenx was the chosen kit designed for E. coli, but…not only for E. coli. Other species were present. Has the authors described the limitations on this test? Or have they checked with the company? Cause all these look like a contamination problem rather that the test problem. Not to mention some unidentified species present as well.
Line 384-395. Shrimp samples were taken after removal of carapace, and the extraction intestines aseptically. Both those were checked for E.coli and from which E. coli was found. If were from shrimp or from intestines makes a big difference. Authors don’t discuss about that.
Also the sampling sites A, B and C are connected to the same River. There are little information regarding these sites. Do they use water from the same river for their aquaculture or they use from pipeline. The majority of such aquacultures will use the river water and after redirect it to the river again. If that is the case here, the authors should present how one affects the other aquaculture and discuss about it. The presence of 3 aquacultures emptying all water to the same river means that the authors should have taken water samples from river water after the 3rd aquaculture and before city begins and compare it with the ones from aquaculture.
Author Response
Reviewer #3
The article is well written but there are some key elements that need to be clarified since there is some confusion with the sampling process and the presentation of the results. In table 1, how many times was A sampled? And why not the same for B and C?
In Table 1, sample location A had a total of 54 water samples tested (18 from water influent, 18 from effluent water and 18 from the shrimp pond). The same number of samples were tested at sample location B and C.
In Table 2 there is a difference in results between A and B-C, has this been explained in article? Because their water samples are well infected
This is a really useful comment and we have changed Table 2 to reflect the challenge of quantifying CFUs in the shrimp samples. During the study, we often identified growth in the shrimp gut samples, however, there were some biochemical reactions and color changes within the test kits that affected our ability to clearly count the CFUs in the test kits. We have now dropped the colony forming units from Table 2 in order to address this point, and we have added new text that describes the difficulty we faced in quantifying E. coli directly from the shrimp gut samples.
Line 117-120. WGS was conducted to a “random” subset of isolates. Why was that? And why the majority was chosen from water samples 34 and only 10 from shrimp samples?
We had limited resources for whole genome sequencing, and isolates were selected from each media type, water or shrimp, at numbers that corresponded to the number of isolates from that media that we identified. Given that shrimp is likely to be consumed by humans and could be an important source of human exposure to ABR bacteria, we agree that more focus should have been on sequencing isolates from shrimp.
Line142-148. Aquagenx was the chosen kit designed for E. coli, but…not only for E. coli. Other species were present. Has the authors described the limitations on this test? Or have they checked with the company? Cause all these look like a contamination problem rather that the test problem. Not to mention some unidentified species present as well.
We have now added more about the limitations to the methods used in this study. We chose the Aquagenx GEL EC kit, which detects and quantifies E. coli based on enzyme-substrate reactions from water samples.The media used by Aquagenx (Aquagenx, Chapel Hill, NC), uses a β-D-glucuronide E. coli indicator. This media can sometimes support the growth of a broader set of bacteria beyond E. coli, especially when colony forming unit counts are high. According to a report from the company: “The Aquagenx Gel was also assessed for false positives by using concentrated stocks of six non-target bacteria (Aeromonas, Citrobacter, Enterobacter, Klebsiella, Pseudomonas aeruginosa, and Serratia). The Aquagenx Gel did not report any false positive values in the absence of E. coli.”
Despite the Aquagenx GEL EC kit being designed for E. coli, it has some limitations, including potential cross-reactivity, non-specific detection, and challenges in interpreting results when other bacterial species are present. Field controls, including both positive and negative, were used to ensure quality control and ensure that contamination was not an issue. In one study of 51 water samples that tested positive for E. coli by the Aquagenx, kit, indole testing biochemically confirmed 19 samples to contain E. coli (Rayasam et al. 2019).
Rayasam, S.D., Ray, I., Smith, K.R. and Riley, L.W., 2019. Extraintestinal pathogenic escherichia coli and antimicrobial drug resistance in a maharashtrian drinking water system. The American journal of tropical medicine and hygiene, 100(5), p.1101.
Line 384-395. Shrimp samples were taken after removal of carapace, and the extraction intestines aseptically. Both those were checked for E. coli and from which E. coli was found. If were from shrimp or from intestines makes a big difference. Authors don’t discuss about that.
In pre-testing our methods prior to the study, we tried rinsing whole shrimps and then culturing the rinsate from the whole shrimp. However, there were biochemical reactions that occurred in the sample kits that prevented us from being able to read the results. Once we removed the carapace of the shrimp and used the shrimp intestines, we were able to reduce the biochemical reaction and read the results more clearly. We have added new text to describe this issue in more detail.
Also the sampling sites A, B and C are connected to the same River. There are little information regarding these sites. Do they use water from the same river for their aquaculture or they use from pipeline. The majority of such aquacultures will use the river water and after redirect it to the river again. If that is the case here, the authors should present how one affects the other aquaculture and discuss about it. The presence of 3 aquacultures emptying all water to the same river means that the authors should have taken water samples from river water after the 3rd aquaculture and before city begins and compare it with the ones from aquaculture.
We have added information about how the shrimp farms connect to the Atacames river. As noted in the comment, all of the shrimp farms are connected to same river via channels, and the effluent from each of the farms is also connected to the Atacames river via channels. We only had access to collect samples from where water came into the pond from the Atacames river and the effluent points where water exited the pond. We were unable to follow the effluent channels to where they enter back into the river.
Reviewer 4 Report
Comments and Suggestions for Authors
This manuscript describes the presence of antibiotic resistance genes (ARGs) in shrimp and water from shrimp farms in Ecuador. It was demonstrated that ARGs of medically important antibiotics are very common in the study area, thereby posing a very serious threat to general public health. This information is useful for regulatory authorities to regulate antibiotic overuse and misuse in shrimp aquaculture. The manuscript writing is generally good, and the benefit to readers is obvious. Yet, some minor issues needed to be clarified or improved as follows:
1. Many drugs can be classified as third-generation cephalosporins (not just only ceftriaxone). However, the authors used only ceftriaxone in this study. To avoid any confusion, I strongly suggest that the word “third-generation cephalosporin” or “3GC” should be replaced by “ceftriaxone” throughout the manuscript.
2. The authors mentioned “Extended-spectrum β-lactamases (ESBL) genes” in the abstract, but the information was not available in the Results. Therefore, please indicate the detected ESBL genes in the Results.
3. Line 87: What are differences between the farms A, B, and C in terms of environmental factors or farm management?
4. Line 89: Please specify what was the “shrimp sample” used for bacterial isolation (such as hepatopancreas, intestine, or hemolymph).
5. Line 100-101: It was unclear why the authors focused mainly on ceftriaxone in this study. Does it now become the most frequently used antimicrobial drug in Ecuadorian shrimp farming or is it a serious issue in Ecuador recently? Please explain more.
6. Line 101: Please indicate the final concentration of ceftriaxone added to the media and briefly explain why this concentration was chosen by the authors.
7. Line 121-137: Please specify the detected ARGs for each antibiotic. For example, tet A and tet B genes for tetracyclines.
8. Line 141-148 and Fig 3: Surprisingly, Vibrio spp. was apparently absent from the water and shrimp samples despite this genus is generally ubiquitous in marine and brackish water environments including shrimp farms. Please discuss this finding.
9. Figure 4: Please consider presenting the ARGs results by using a stacked bar chart instead of the simple bar chart such that the relative abundant of ARGs in the water and shrimp samples can be seen.
10. Line 249: Please write the full name of “P. alimentorum” when first mentioned.
11. Line 371-376: Please provide the water quality data of the shrimp pond water, especially water salinity and temperature.
Author Response
Reviewer #4
This manuscript describes the presence of antibiotic resistance genes (ARGs) in shrimp and water from shrimp farms in Ecuador. It was demonstrated that ARGs of medically important antibiotics are very common in the study area, thereby posing a very serious threat to general public health. This information is useful for regulatory authorities to regulate antibiotic overuse and misuse in shrimp aquaculture. The manuscript writing is generally good, and the benefit to readers is obvious. Yet, some minor issues needed to be clarified or improved as follows:
- Many drugs can be classified as third-generation cephalosporins (not just only ceftriaxone). However, the authors used only ceftriaxone in this study. To avoid any confusion, I strongly suggest that the word “third-generation cephalosporin” or “3GC” should be replaced by “ceftriaxone” throughout the manuscript.
We have now changed this throughout the manuscript.
- The authors mentioned “Extended-spectrum β-lactamases (ESBL) genes” in the abstract, but the information was not available in the Results. Therefore, please indicate the detected ESBL genes in the Results.
We have added new text about the specific ESBL genes identified, and they are now in the gene list (see line 124).
- Line 87: What are differences between the farms A, B, and C in terms of environmental factors or farm management?
We have added additional information about the farms, which was limited due to safety concerns in this region when the study was being conducted. This is an important question, however, we were not able to get farm management information from the shrimp production operators about antibiotic use or other farm management practices. All of the farms received water from the Atacames river, prior to its arrival to the city of Atacames.
- Line 89: Please specify what was the “shrimp sample” used for bacterial isolation (such as hepatopancreas, intestine, or hemolymph).
Each shrimp sample involved the aseptic removal of the carapace and the extraction of the shrimp intestine. The isolated shrimp intestine was then placed in a whirl pack bag along with 100mL of autoclaved distilled water and incubated in the same Aquagenx kit used for water samples.
- Line 100-101: It was unclear why the authors focused mainly on ceftriaxone in this study. Does it now become the most frequently used antimicrobial drug in Ecuadorian shrimp farming or is it a serious issue in Ecuador recently? Please explain more.
Ceftriaxone is a third-generation cephalosporin frequently used to treat severe infections caused by E. coli. Additionally, resistance to ceftriaxone is often mediated by extended-spectrum beta-lactamases (ESBLs). These enzymes not only inactivate third-generation cephalosporins but also other beta-lactam antibiotics, further limiting available therapeutic options. Many previous studies have used ceftriaxone to select for ESBL-producing strains (see some examples below).
- Line 101: Please indicate the final concentration of ceftriaxone added to the media and briefly explain why this concentration was chosen by the authors.
The chosen concentration (1 mg/L), while low, reflects environmental levels that might be found in such settings due to anthropogenic activities or improper disposal of antimicrobials. Moreover, this concentration is commonly used in studies to select strains of E. coli resistant to this cephalosporin (Chirindze et al., 2018; Landolsi et al., 2022; Johansson et al., 2022; Aldea et al., 2022).
Chirindze, L.M., Zimba, T.F., Sekyere, J.O. et al. Faecal colonization of E. coli and Klebsiella spp. producing extended-spectrum beta-lactamases and plasmid-mediated AmpC in Mozambican university students. BMC Infect Dis 18, 244 (2018). https://doi.org/10.1186/s12879-018-3154-1
Landolsi, S.; Selmi, R.; Hadjadj, L.; Ben Haj Yahia, A.; Ben Romdhane, K.; Messadi, L.; Rolain, J.M. First Report of Extended-Spectrum beta-Lactamase (blaCTX-M1) and Colistin Resistance Gene mcr-1 in E. coli of Lineage ST648 from Cockroaches in Tunisia. Microbiol. Spectr. 2022, 10, e0003621.
Johansson, V.; Nykasenoja, S.; Myllyniemi, A.L.; Rossow, H.; Heikinheimo, A. Genomic characterization of ESBL/AmpCproducing and high-risk clonal lineages of Escherichia coli and Klebsiella pneumoniae in imported dogs with shelter and stray background. J. Glob. Antimicrob. Resist. 2022, 30, 183–190.
Aldea, I.; Gibello, A.; Hernandez, M.; Leekitcharoenphon, P.; Bortolaia, V.; Moreno, M.A. Clonal and plasmid mediated flow of ESBL/AmpC genes in Escherichia coli in a commercial laying hen farm. Vet. Microbiol. 2022, 270, 109453.
- Line 121-137: Please specify the detected ARGs for each antibiotic. For example, tet A and tet B genes for tetracyclines.
Thank you for the feedback, manuscript now updated.
- Line 141-148 and Fig 3: Surprisingly, Vibrio spp. was apparently absent from the water and shrimp samples despite this genus is generally ubiquitous in marine and brackish water environments including shrimp farms. Please discuss this finding.
Good point and we have added this to the discussion. It is likely that Vibrio spp. did not have the appropriate media and culture conditions to grow in the Aquagenx kits that we used. Our intention with this study was to culture E. coli, however, the media allowed for some other species to grow (Figure 3), which we subsequently realized after whole genome sequencing.
- Figure 4: Please consider presenting the ARGs results by using a stacked bar chart instead of the simple bar chart such that the relative abundant of ARGs in the water and shrimp samples can be seen.
Thank you for the feedback, figure now updated.
- Line 249: Please write the full name of “P. alimentorum” when first mentioned.
Thank you for catching that, fixed now.
- Line 371-376: Please provide the water quality data of the shrimp pond water, especially water salinity and temperature.
We unfortunately did not have the equipment to test other water quality parameters in the ponds (e.g., temperature, dissolved oxygen, salinity, turbidity, etc.), and only bacteria were analyzed for this study.
Round 2
Reviewer 1 Report
Comments and Suggestions for Authors
The authors have addressed my concerns, and the manuscript's quality has significantly improved. I now recommend it for acceptance.
Reviewer 2 Report
Comments and Suggestions for Authors
The authors included in the supplementary file the data confirming the appropriate detection of the ARGs, as was requested.
Comments on the Quality of English LanguageEnglish grammar and wording were improved